# BK channel density is regulated by endoplasmic reticulum associated degradation and influenced by the SKN-1A/NRF1 transcription factor

**Timothy P. Cheung**[1,2], **Jun-Yong Choe**[2,3], **Janet E. Richmond**[4], **Hongkyun Kim**[1,2]*

**1** Center for Cancer Cell Biology, Immunology, and Infection, Department of Cell Biology and Anatomy, Chicago Medical School, Rosalind Franklin University of Medicine & Science, North Chicago, Illinois, United States of America, **2** School of Graduate & Postdoctoral Studies, Rosalind Franklin University of Medicine & Science, North Chicago, Illinois, United States of America, **3** Department of Biochemistry and Molecular Biology, Rosalind Franklin University of Medicine & Science, North Chicago, Illinois United States of America, **4** Department of Biological Sciences, University of Illinois at Chicago, Chicago, Illinois, United States of America

* hongkyun.kim@rosalindfranklin.edu

**Data Availability Statement:** All relevant data are within the manuscript and its Supporting Information files.

## Abstract

Ion channels are present at specific levels within subcellular compartments of excitable cells. The regulation of ion channel trafficking and targeting is an effective way to control cell excitability. The BK channel is a calcium-activated potassium channel that serves as a negative feedback mechanism at presynaptic axon terminals and sites of muscle excitation. The *C. elegans* BK channel ortholog, SLO-1, requires an endoplasmic reticulum (ER) membrane protein for efficient anterograde transport to these locations. Here, we found that, in the absence of this ER membrane protein, SLO-1 channels that are seemingly normally folded and expressed at physiological levels undergo SEL-11/HRD1-mediated ER-associated degradation (ERAD). This SLO-1 degradation is also indirectly regulated by a SKN-1A/NRF1-mediated transcriptional mechanism that controls proteasome levels. Therefore, our data indicate that SLO-1 channel density is regulated by the competitive balance between the efficiency of ER trafficking machinery and the capacity of ERAD.

## Author summary

Excitable cells, such as neurons and muscles, are essential for the movement and behavior of animals. These cells express a set of specific types of ion channels that allow the selective passage of ions across the plasma membrane. The alteration in the levels of these ion channels influences cell excitability and the function of excitable cells. The regulation of ion channel trafficking and targeting is an effective way to control the function of excitable cells. The BK SLO-1 channel is a calcium-activated potassium channel that reduces excitability at presynaptic axon terminals and sites of muscle excitation. In a *C. elegans* genetic study, authors found that the delayed exit of SLO-1 channels from the ER causes their

**Funding:** HK and JR received funding from R01GM125749 grant National Institute of Health <https://www.nih.gov/grants-funding>. JC received funding from R01GM123103 grant National Institute of Health <https://www.nih.gov/grants-funding>. The funders had no role in study design, data collection and analysis, decision to publish, or preparation of the manuscript.

**Competing interests:** The authors have declared that no competing interests exist.

degradation by a mechanism called ER-associated degradation (ERAD). Interestingly, the same components that directly mediate SLO-1 ERAD also process a key transcriptional factor that maintains proteasome levels, thus indirectly influencing SLO-1 degradation. These data show that the levels of SLO-1 channels are regulated by the competitive balance between the efficiency of ER trafficking machinery and the capacity of ERAD.

## Introduction

The BK (KCa1.1, MaxiK, SLO1) channel is a large conductance calcium- and voltage-activated $K^+$ channel, widely expressed in a variety of cell types. Its myriad of functions include regulation: of neuronal excitability, synaptic transmission, muscle excitation, vascular tone, renal secretion, and hormone release [1–8]. Disruptions in BK channel function and distribution are associated with various diseases, ranging from hypertension [9–11], alcoholism [12,13], and epilepsy [14], to neurodegenerative diseases such as Fragile X syndrome [15] and Alzheimer's Disease [16].

The BK channel, a tetramer of pore-forming α subunits provides a negative feedback mechanism in response to membrane-depolarizing stimuli and increased cytosolic $Ca^{2+}$ levels, thus preventing the excessive release of neurotransmitters from the presynaptic neuron or hyperexcitability in the muscle (Reviewed in [17]). The α subunit consists of seven transmembrane segments, which form the voltage-sensor and pore-gating domains, and a long cytoplasmic tail, which triggers allosteric conformational changes that open the pore by binding $Ca^{2+}$ via two $Ca^{2+}$-binding sites within the RCK1 (regulator of conductance for $K^+$) and RCK2 domains [18,19].

The unique sensitivity of BK channels to $Ca^{2+}$ highlights the importance of channel trafficking and localization since $Ca^{2+}$ influx in the excitable cells is spatially restricted to specialized compartments, termed calcium nanodomains. However, the molecular mechanisms underlying BK channel trafficking and localization are unclear. In previous *C. elegans* genetic studies, we identified genes required for the localization of SLO-1, the BK channel ortholog, to the active zones of presynaptic terminals and muscle excitation sites [20,21]. Interestingly, mutations in these genes caused diffuse SLO-1 channel localization but did not alter the overall level of SLO-1 channels, indicating that endocytosis at the plasma membrane is not a major regulatory mechanism for SLO-1 channel density. Recently, we found that ERG-28, an ER membrane protein, plays a crucial role in the anterograde trafficking of SLO-1 from the ER to the Golgi complex. ERG-28 contains a cytosolic ER retention/retrieval motif (KKXX-COOH), which is associated with coat-protein complex I (COPI)-related retrograde trafficking vesicles [22,23] and is essential for SLO-1 trafficking [21]. A null mutation in the *erg-28* gene significantly reduces SLO-1 delivery to presynaptic terminals and muscle excitation sites. Importantly, SLO-1 levels are reduced throughout the entire animal, indicating that SLO-1 channel density is largely determined by the anterograde trafficking process from the ER to the Golgi complex. This study further suggests that SLO-1 channels are degraded by ER-associated degradation (ERAD) rather than sequestered to another intracellular compartment. However, ERAD does not typically target normally folded proteins in the absence of ER stress. Currently, known exceptions are found in only two sterol biosynthetic enzymes, whose degradation is regulated by specific signals [24–26].

Ubiquitinated ERAD substrates are transferred from the ER to the proteasome for degradation. Proteasome dysfunction induces the expression of proteasome subunit genes through the Cap 'n' Collar (CnC) bZIP transcription factor NRF1. Its *C. elegans* homolog, SKN-1A, is one

of the three isoforms generated by the single *skn-1* gene. Unlike the other isoforms, SKN-1A is targeted to the ER, where it is glycosylated and normally undergoes degradation by the SEL-11-dependent ERAD pathway. Upon proteasome dysfunction, SKN-1A escapes from ERAD and becomes activated through a cascade of events in which it is deglycosylated and sequence-edited by PNG-1/NGLY1 (peptide:N-glycanase 1), translocated to the nucleus, and cleaved by the aspartic protease DDI-1/DDI2 [27–29]. The activated SKN-1A (hereafter referred to as SKN-1A[cut, 4ND]) upregulates the expression of the proteasome subunit genes.

Here, we report that SLO-1 channels are degraded by SEL-11/HRD1-mediated ERAD. Also, we found that the SLO-1 degradation pathway overlaps with the SKN-1A-mediated proteasome induction pathway. Therefore, the lack of these components or enzymes affects the density of functional SLO-1 levels at neurons and muscles by directly reducing SLO-1 degradation and indirectly by SKN-1A activation.

## Results

### The SEL-11/HRD1 E3 ubiquitin ligase participates in the ERAD of functional SLO-1 channels

In a previous genetic study, we showed that a loss-of-function mutation in *erg-28*, a gene encoding an ER membrane protein, resulted in a marked reduction of SLO-1 channels in the presynaptic terminals and muscle excitation sites [21]. Because ERG-28 is a protein shuttling between the ER and the pre-Golgi intermediate compartment, we hypothesized that inefficient trafficking of SLO-1 channels in *erg-28* mutants causes channel degradation by ER-associated degradation (ERAD). A key protein that mediates ERAD is an E3 ubiquitin ligase present in the ER membrane. To test this hypothesis, we introduced null mutations of all the known ER resident E3 ubiquitin ligases into *erg-28(gk697770) slo-1(cim105[slo-1*::*GFP])* animals (S1 Fig). The E3 ubiquitin ligases tested were chosen based on literature and included *rnf-5*, *rnf-121*, *hrdl-1*, *marc-6*, and *sel-11* [30–34]. If a specific E3 ubiquitin ligase is necessary for the ubiquitination and degradation of SLO-1 in *erg-28* mutants, then its deletion should restore SLO-1 levels in the dorsal cords and body wall muscles. Among the mutations we tested, we found that *sel-11(tm1743)*, which has a deletion in the gene encoding a homolog of HRD1 (HMG-CoA reductase degradation) E3 ubiquitin ligase, significantly elevated SLO-1 channel levels in the *erg-28* mutant background (Fig 1A and 1B). To exclude the possibility that the *sel-11* mutation increased the levels of SLO-1 channels independently of *erg-28* mutation, we tested the levels of SLO-1 channels in *sel-11* mutant animals but did not find a significant change. Together, these results support the idea that the delayed trafficking of SLO-1 channels in the absence of ERG-28 causes their degradation in a SEL-11 dependent manner (Fig 1A and 1B).

To further confirm our findings that SEL-11 is the major E3 ubiquitin ligase responsible for SLO-1 degradation in *erg-28* mutant animals, we measured SLO-1 levels using Western blot analysis. Consistent with our *in vivo* fluorescent imaging data, we found that the reduced SLO-1 channel protein level in *erg-28* mutant animals was considerably restored in *sel-11 erg-28* double mutant animals (Fig 1C). Together, our results indicate that the delayed trafficking of SLO-1 channels in the absence of ERG-28 renders them susceptible to SEL-11-dependent ERAD.

SEL-11 is a known component of the endoplasmic reticulum unfolded protein response (UPR$^{ER}$), a critical homeostatic mechanism that alleviates ER stress. It is possible that the reduction of SLO-1 channels in *erg-28* mutant animals results from ER stress caused by the absence of ERG-28. The expression of the *hsp-4 (*grp78/BiP ortholog) gene is upregulated during ER stress, and its promoter-tagged GFP reporter has been successfully used to measure ER stress [32,33,35]. We found that the *erg-28* mutation did not induce the expression of the *hsp-*

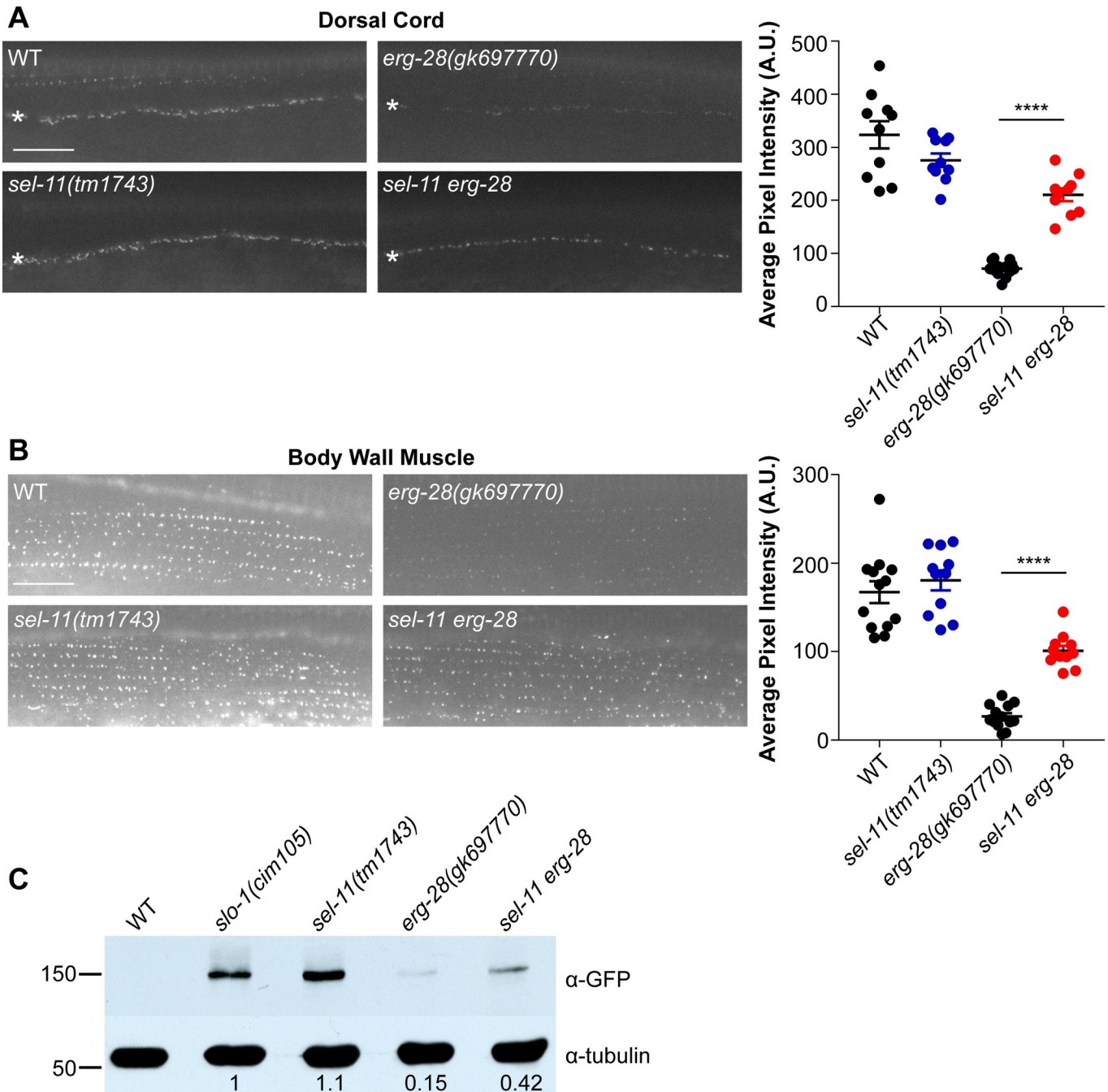

**Fig 1. SEL-11/HRD1 is required for normal degradation of SLO-1. (A)** and **(B)** Representative images and quantification of SLO-1 in the dorsal cord and body wall muscle (white asterisk indicates dorsal cord). Data are means ± SEM; ****P < 0.0001, One-way ANOVA, Tukey's post hoc test. (scale bar = 10 μm). **(C)** Western blot analysis shows that total SLO-1 levels are reduced in *erg-28* mutants and partially restored in *sel-11 erg-28* double mutants. SLO-1 intensity was normalized to that of the α-tubulin. N2 does not express GFP and is a negative control for western blotting with anti-GFP antibody.

*4*::*gfp* reporter (S2 Fig). We also tested whether *erg-28* is necessary for mediating the ER stress response. If that is the case, *hsp-4*::GFP induction will be attenuated in *erg-28* mutant animals when treated with the known ER stress inducer tunicamycin. We found that *hsp-4*::GFP induction by tunicamycin was not altered in *erg-28* mutant animals, indicating that the reduction of

SLO-1 levels in *erg-28* mutant animals does not involve the UPR$^{ER}$ (S2 Fig). Collectively, these data highlight the specificity of *erg-28* in SLO-1 channel degradation.

Next, we investigated whether the restored SLO-1 channels in *sel-11 erg-28* double mutant animals were functional. For this, we used the *slo-1(ky399)* gain-of-function mutant (hereafter referred to as *slo-1(ky399gf)*), which exhibits sluggish movement due to delayed closing kinetics of SLO-1 channels. As previously reported, the sluggish locomotory speed of *slo-1(ky399gf)* animals was significantly increased by the *erg-28(gk697770)* mutation (Fig 2A). Furthermore, introducing a *sel-11(tm1743)* mutation to *erg-28 slo-1(ky399gf)* animals markedly reduced the locomotory speed, compared to *erg-28 slo-1(ky399gf)* animals (Fig 2A). We previously reported that the sluggish movement of *slo-1(ky399gf)* animals was due to a neuronal presynaptic defect, and *erg-28* functions in neurons to modify the speed of *slo-1(ky399gf)* animals [21]. Hence, we assessed presynaptic function using aldicarb, an acetylcholinesterase inhibitor. The aldicarb-induced paralysis assay is useful in determining presynaptic functional defects, since the time required for the paralysis of a specific mutant animal inversely correlates with the level of its cholinergic neurotransmission [36,37]. As expected, sluggish *slo-1(ky399gf)* mutants showed aldicarb resistance, while *erg-28 slo-1(ky399gf)* double mutants were hypersensitive to aldicarb. Furthermore, the introduction of the *sel-11(tm1743)* mutation to *erg-28 slo-1(ky399gf)* mutants restored aldicarb resistance (Fig 2B).

To further substantiate the aldicarb results, we performed electrophysiological recordings at *C. elegans* neuromuscular junctions. As expected, the evoked amplitude response of *slo-1 (ky399)* animals was reduced compared to wild-type animals, but elevated by the *erg-28 (gk697770)* mutation. The introduction of the *sel-11(tm1743)* mutation to *erg-28 slo-1(ky399gf)* mutant animals depressed the evoked amplitude response (Fig 2C). Together, these data show that the restored SLO-1 channels were functional.

Our data indicate that reduced SLO-1(gf) channel function in the *erg-28* mutant is suppressed by *sel-11* mutation. We examined whether normal SLO-1 channels are similarly affected by *erg-28* and *sel-11* mutations. Previously, we showed that *erg-28* mutants exhibit *slo-1* reduction-of-function phenotypes, including aldicarb hyper-sensitivity [21]. We found that the aldicarb sensitivity of the *erg-28(gk697770)* mutant was considerably suppressed by the *sel-11(tm1743)* mutation (S3 Fig), indicating that restored endogenous SLO-1 channels were fully functional. These results also strongly suggest that SLO-1 channels in the *erg-28* mutant are not appreciably misfolded since *sel-11* mutation could restore their function.

As another way to assess whether SLO-1::GFP is misfolded, we used an integrated transgene that drives SLO-1::GFP expression in cholinergic DA and DB motor neurons to determine the amount of SLO-1::GFP at presynaptic terminals. We hypothesized that if *erg-28* is required for proper folding of SLO-1, then the overexpression of SLO-1 in *erg-28* mutants will prevent further SLO-1 trafficking beyond the ER, resulting in lower levels of SLO-1 at the dorsal cord. The introduction of an *erg-28* mutation did not alter SLO-1 levels at the dorsal cord, consistent with the idea that the degraded SLO-1 were not misfolded (S4 Fig).

## A forward genetic screen identifies *sel-11* as a gene involved in SLO-1 degradation

Our data indicate that ERG-28 is an ER membrane protein specific to SLO-1, and lack thereof leads to SLO-1 degradation by a SEL-11/HRD1-dependent ERAD pathway. We sought to identify additional components of SLO-1 degradation in *erg-28* mutants. Using CRISPR/Cas9 genome editing, we introduced a gain-of-function point mutation in *slo-1(cim105)* animals, which changes the glutamate to glutamine at amino acid position 350. The glutamate residue is positioned at the entrance of the inner vestibule of the SLO-1 channel, and this change is

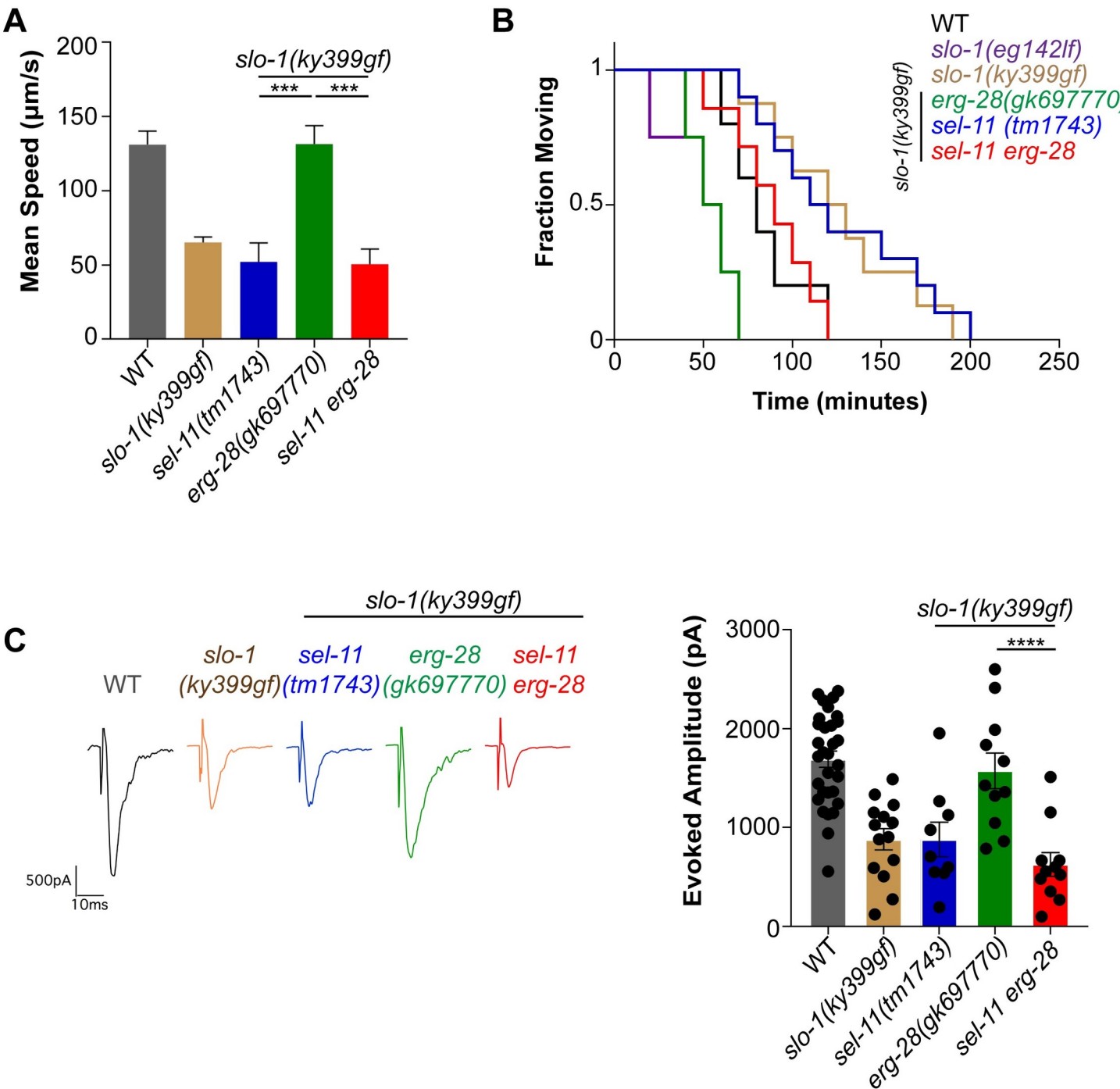

**Fig 2. A *sel-11* mutation reverses the reduced SLO-1 functions observed in the absence of ERG-28. (A)** Locomotory speed was measured in mutants with *slo-1(ky399gf)* backgrounds. *erg-28 slo-1(ky399gf)* mutant animals move faster than *slo-1(ky399gf)*. *sel-11* mutation depresses locomotory speed in *erg-28 slo-1(ky399gf)* mutant animals. Data are means ± SEM; ***P < 0.001, One-way ANOVA, Tukey's post hoc test. **(B)** *sel-11* mutation increases aldicarb resistance in *erg-28 slo-1(ky399gf)* animals. Aldicarb-induced paralysis was analyzed using Kaplan-Meier survival analysis. **(C)** Electrophysiology reveals that a *sel-11* mutation depresses evoked amplitude in *erg-28 slo-1(ky399gf)* mutants. Data are means ± SEM; ****P < 0.0001, One-way ANOVA, Tukey's post hoc test.

known to delay the closing kinetics in mammalian and *C. elegans* BK channels [38]. The resulting strain, *slo-1(cim113[slo-1E350Q]::GFP)*, exhibited a visually sluggish and slow locomotory phenotype, yet retained similar levels of SLO-1 when compared to *slo-1(cim105)* (Fig 3A).

*C. elegans* exhibits high-frequency lateral swimming ("thrashing") behavior in a liquid medium, and the measurement of its frequency is a well-established method for assessing energetically challenging mobility. Compared to wild-type animals, *slo-1(cim113gf)* animals exhibited low-frequency thrashing (Fig 3E). An *erg-28* mutation increased the thrash rate in *slo-1 (cim113gf)* animals and drastically reduced SLO-1, to levels that were unquantifiable with fluorescent microscopy (Fig 3B).

To identify additional components involved in SLO-1 channel degradation, we performed a forward genetic screen using ethyl methanesulfonate (EMS) with *erg-28 slo-1(cim113gf)* mutant animals. We first isolated candidate suppressor animals that exhibited sluggish movements with a low thrashing frequency in F2 progeny of *erg-28 slo-1(cim113gf)* mutant animals. Then, we selected mutants with elevated levels of SLO-1 in the nerve ring. The confirmed mutant animals were subjected to whole-genome sequencing. One of the mutants, *cim54*, has a missense mutation in the *sel-11* gene, in which proline in the 371 amino acid position is substituted for a serine. This proline is located within the highly conserved proline-rich region, which is proposed to recognize substrates for ubiquitination (Fig 3G) [39]. The *sel-11(cim54)* mutation increased SLO-1 levels and reduced thrashing rates in *erg-28 slo-1(cim113gf)* animals (Fig 3D–3F). Moreover, a transgenic *sel-11(cim54) erg-28 slo-1(cim113gf)* line carrying a fosmid that contains a wild-type copy of *sel-11* exhibits increased thrash rates and reduced SLO-1 levels, comparable to those of *erg-28 slo-1(cim113gf)* animals (Fig 3D–3F). Interestingly, the introduction of a *sel-11(tm1743)* deletion mutant animals exhibited higher levels of SLO-1 than *sel-11(cim54)* missense mutants, in *erg-28 slo-1(cim113gf)* backgrounds, suggesting that *cim54* is not a null mutation (S5 Fig). Together, the identification of SEL-11 as a mediator of SLO-1 degradation in *erg-28* mutants, through two independent experiments, including a candidate gene approach of ER-resident E3 ubiquitin ligases and an unbiased forward genetic screen, establishes SEL-11 as a central component of SLO-1 degradation.

## SEL-11/HRD1-associated ERAD proteins participate in SLO-1 degradation

The identification of SEL-11-mediated degradation as a central component of the SLO-1 ERAD pathway prompted us to investigate the participation of other known SEL-11- associated ERAD components. SEL-11/HRD1 forms a complex with SEL-1/HRD3, which is required for substrate recognition, degradation, and stability of the HRD1 E3 ligase [40–42]. To determine whether SEL-1 is necessary for SLO-1 degradation, we introduced a nonsense mutation into the *sel-1* gene of *erg-28 slo-1(cim105)* using CRISPR/Cas9 genome-editing technology. *sel-1(cim115) erg-28 slo-1(cim105)* mutant animals exhibited higher levels of SLO-1 channels in both neurons and muscles than *erg-28 slo-1(cim105)* animals (Fig 4A and 4B), implicating SEL-1 in SEL-11-mediated SLO-1 degradation.

Another component of the SEL-11/HRD1 complex is Derlin. Although the exact role of Derlin in ERAD is not clearly defined, Derlin associates with HRD1 through a scaffolding protein Usa1 and is essential for ERAD [43]. Derlin shares significant homology with rhomboid family intramembrane proteases but has no enzymatic activity. For this reason, Derlin has been proposed to recognize substrates and transfer them to HRD1 for ERAD [44–47]. While mammals have three different Derlin proteins (Der1, Der2 and Der3), *C. elegans* possesses two Derlin homologs *cup-2* and *der-2*. To determine whether the two Derlin homologs participate in SLO-1 degradation, we introduced the null mutations of *cup-2(tm2909)* and *der-2(tm6098)*, into *erg-28* mutant animals. Interestingly, neither *cup-2* nor *der-2* mutation altered SLO-1 levels (S6A and S6B Fig). To test the possibility of functional redundancy within the Derlin homologs in SLO-1 degradation, we introduced both *cup-2* and *der-2* mutations into *erg-28* mutant animals. The simultaneous mutations of both Derlin homologs significantly restored

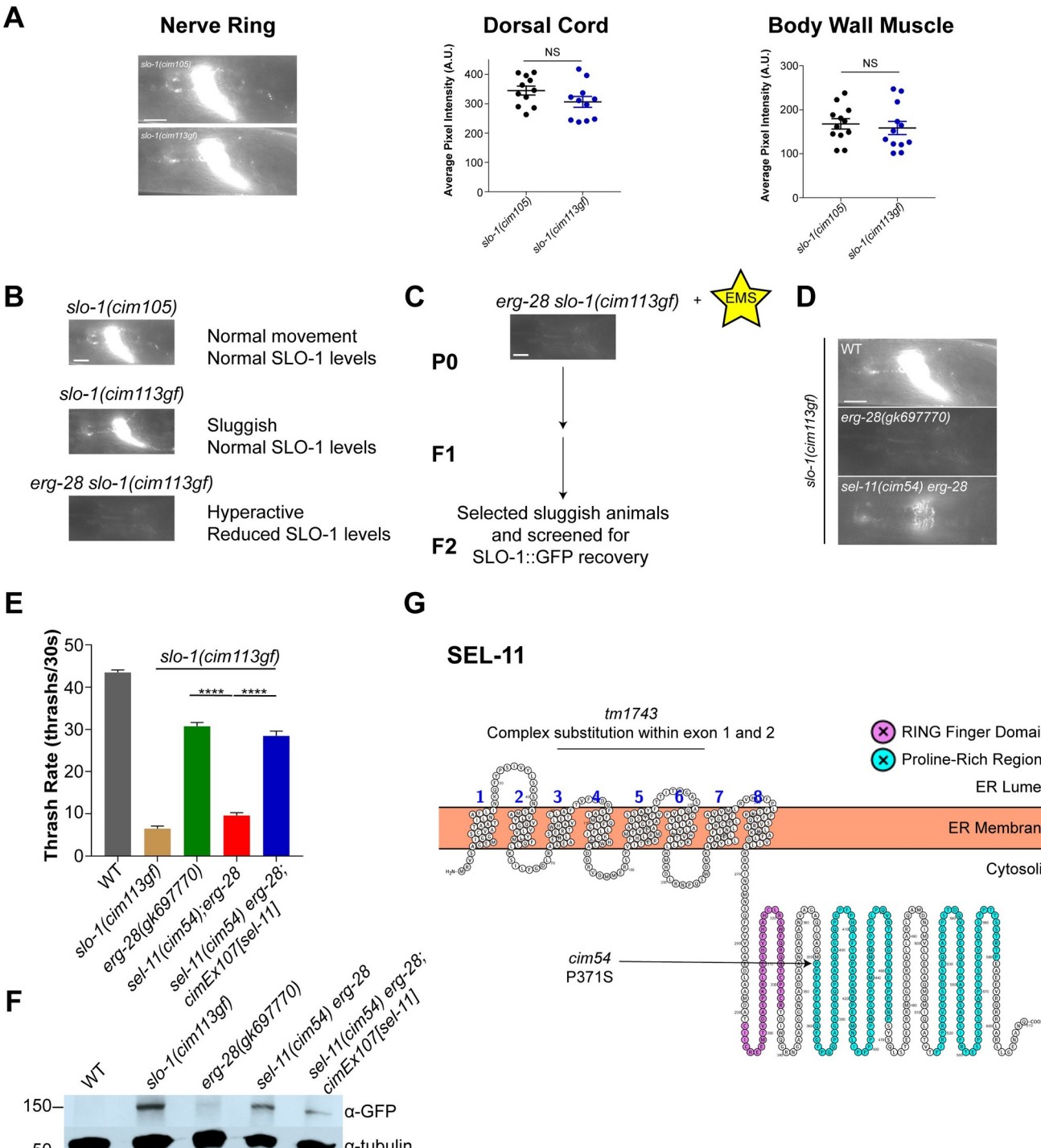

**Fig 3. A forward genetic screen reveals *sel-11* as a critical gene for SLO-1 degradation. (A)** Representative images of SLO-1 levels in *slo-1(cim105)* and *slo-1(cim113gf)* nerve rings. (scale bar = 10 µm). SLO-1 levels of the dorsal cords and body wall muscles were similar between *cim105* and *cim113gf*. Data are means ± SEM; NS, paired two-tailed t-test. **(B)** Phenotypic changes and the associated SLO-1 levels. **(C)** Schematic of the EMS forward genetic screen. **(D)** Representative image of the nerve ring of *cim54*, a mutant isolated from the screen. **(E)** *sel-11(cim54)* mutation abolished the thrash rate recovery by the *erg-28* mutation. *cimEx107*, a transgene expressing a fosmid with the *sel-11* gene, restored the thrash rates in *sel-11(cim54) erg-28 slo-1(cim113gf)* mutants. Data are means ± SEM; ****P < 0.0001, One-way ANOVA, Tukey's post hoc test. **(F)** Western blot analysis shows that total SLO-1 levels were reduced in *erg-28* animals and restored in *sel-11(cim54) erg-28 slo-1(cim113gf)* mutants. The extrachromosomal *cimEx107* transgene reduced SLO-1 levels toward *erg-28 slo-1(cim113gf)* mutants. N2 does not express GFP and is a negative control for Western blotting with anti-GFP antibody. **(G)** Model of SEL-11, based on a previously described report [42], as a multi-pass transmembrane ER protein with a Really Interesting New Gene (RING) finger domain

(purple) and proline-rich regions (teal). The arrow indicates the predicted proline to serine point mutation at position 371, the beginning of the proline-rich regions.The line indicates the predicted complex substitution in the *sel-11(tm1743)* allele.

SLO-1 levels in both neurons and muscles of *erg-28* animals, indicating functional redundancy of CUP-2 and DER-2 in SLO-1 channel degradation (S6A and S6B Fig).

The ERAD system utilizes the p97/VCP/Cdc48 unfoldase to direct ubiquitinated substrates to the 26S proteasome for degradation [48,49]. The Cdc48 unfoldase forms a hexameric assembly that pulls ubiquitinated polypeptide substrates out of the membrane for subsequent

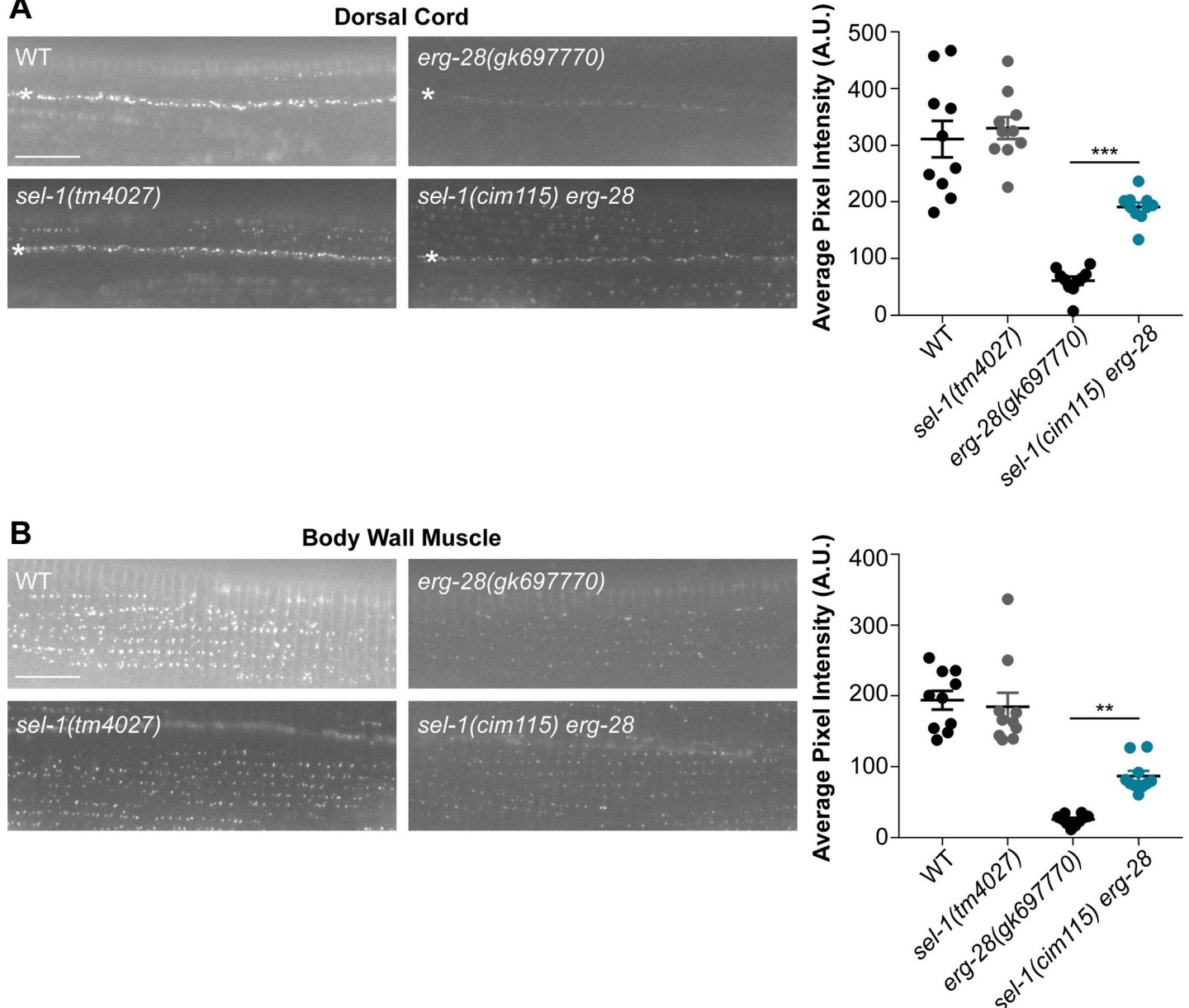

**Fig 4. SEL-1/HRD3, a member of the SEL-11 ubiquitin ligase complex, targets SLO-1 channels for degradation.** (**A**) and (**B**) Representative images and quantification of SLO-1 at the dorsal cord and body wall muscle. (scale bar = 10 μm). Data are means ± SEM; ***P < 0.001; **P < 0.01, One-way ANOVA, Tukey's post hoc test.

delivery to the proteasome. In *C. elegans*, there are two known homologs of Cdc48: *cdc-48.1*, and *cdc-48.2*, both of which have essential and redundant functions [50]. We examined whether these two genes are involved in SLO-1 degradation. We found that introduction of a mutation in *cdc-48.2*, but not *cdc-48.1*, significantly restored SLO-1 levels in both neurons and muscles of *erg-28* mutants (S7A–S7D Fig), indicating that CDC-48.2, but not CDC-48.1, is required for SLO-1 degradation.

## The aspartic protease DDI-1 and SEL-11 work in the same pathway to degrade SLO-1

Previously, we showed that the aspartic protease DDI-1 is involved in SLO-1 degradation in *erg-28* animals [21]. Our current study shows that SEL-11 E3 ubiquitin ligase-dependent ERAD is responsible for SLO-1 degradation. Hence, we sought to elucidate the relationship between SEL-11 and DDI-1. DDI-1 has a conserved aspartic protease domain but also possesses ubiquitin-binding domains [51,52]. To determine whether protease activity is essential for the DDI-1 function, we introduced a *ddi-1(mg572)* mutation, which selectively inactivates the catalytic function of the aspartic protease [27], into *erg-28* mutants, and examined SLO-1 channel levels. We found that the levels of SLO-1 channels in *ddi-1(mg572);erg-28* mutant animals were significantly restored at the presynaptic terminals and muscle excitation sites (Fig 5A). Furthermore, the levels of SLO-1 channels in *ddi-1(mg572);erg-28* mutant animals were similar to those of *ddi-1(ok1468)* null mutant animals (Fig 5A). These results indicate that the aspartic protease activity of DDI-1 is essential for SLO-1 degradation.

Next, we determined whether DDI-1 and SEL-11 work in series in the SLO-1 ERAD pathway. If that is the case, the effect of each mutation on SLO-1 channels would not be additive. Indeed, SLO-1 channel levels in *ddi-1;sel-11 erg-28* mutant animals were comparable with those of *sel-11 erg-28* and *ddi-1;erg-28* animals (Fig 5B). These results indicate that SEL-11 and DDI-1 work sequentially in the same SLO-1 ERAD pathway. Finally, to test whether SEL-11 and DDI-1 work together upstream of the 26S proteasome, we treated the *ddi-1;sel-11 erg-28* animals with bortezomib, a proteasome inhibitor, and found no further increases in SLO-1 levels (S8A and S8B Fig). Altogether, our data suggest that SEL-11 and DDI-1 work in series, upstream of the proteasome, for SLO-1 degradation in *erg-28* mutants.

Finally, we explored the possibility that SLO-1 is degraded by other *sel-11*-independent processes, such as autophagy, because SLO-1 channels are not completely restored to wild-type levels in the *sel-11 erg-28* double mutant. To test whether the autophagy/lysosome pathway contributes to SLO-1 degradation, we introduced an *atg-4.2* mutation into *erg-28* mutant animals. ATG-4.2 is required for autophagosome maturation and enables lysosomal fusion in neurons [53]. If SLO-1 is degraded by autophagy, then the introduction of the *atg-4.2* mutation to *erg-28* animals will elevate SLO-1 levels in the dorsal cord. The *atg-4.2* mutation did not alter SLO-1 levels in *erg-28* mutants (S9 Fig). These results indicate that, while we cannot rule out other degradation mechanisms, including other E3 ubiquitin ligases, SLO-1 is not degraded by autophagy in *erg-28* mutants.

## Proteasomal dysfunction response-mediator, SKN-1A/NRF1, functions in parallel with the classical SEL-11-dependent pathway in SLO-1 degradation

Impaired proteasome function activates the transcription factor SKN-1A, which in turn induces the expression of proteasome subunit genes [27–29,54,55]. ER-associated SKN-1A undergoes elaborate molecular modification and processing steps to become an active transcription factor: retrotranslocation from the ER membrane by SEL-11 and SEL-1, deglycosylation by the peptide N-glycanase PNG-1 [29], and processing by the aspartic protease DDI-1

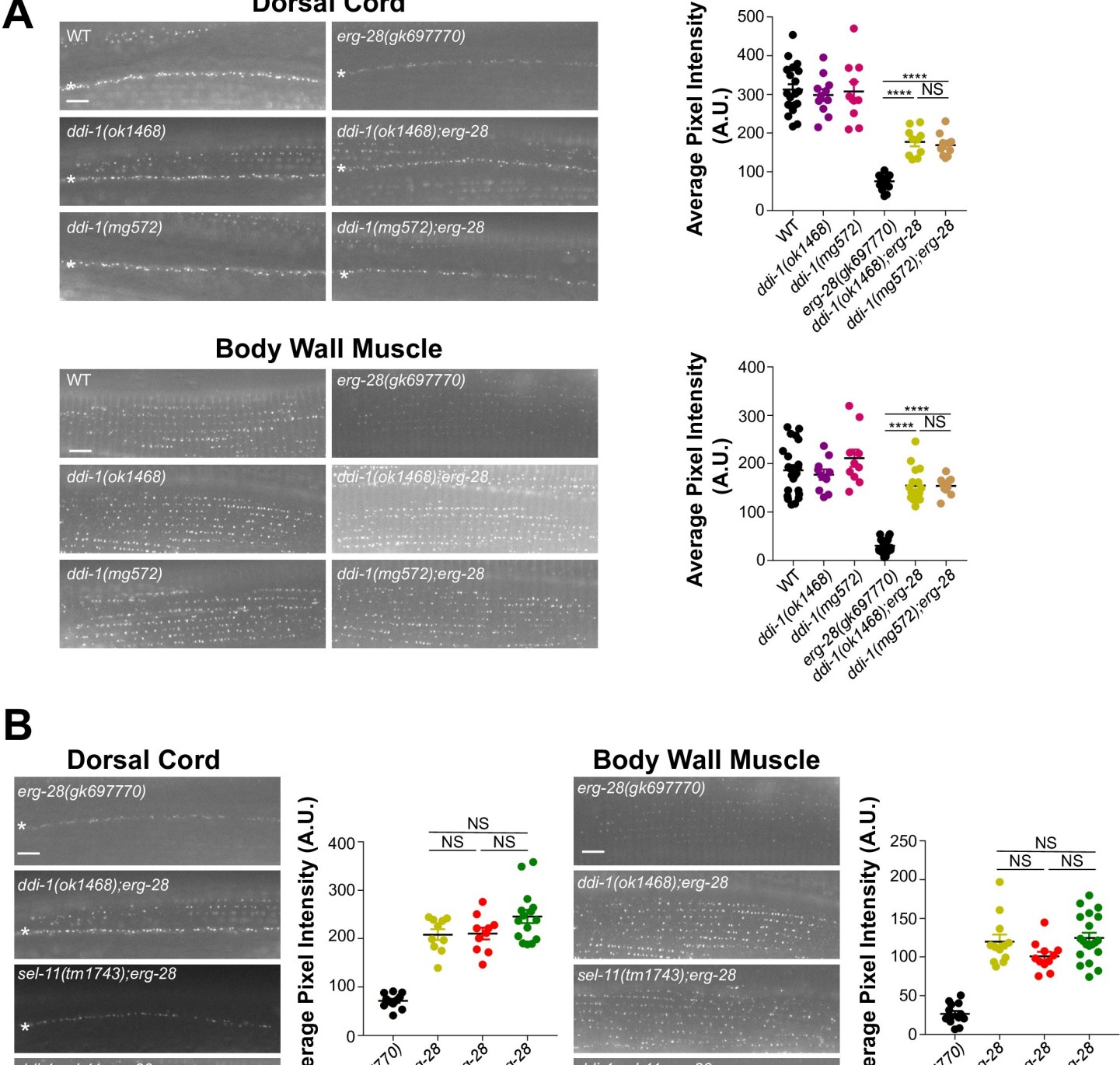

**Fig 5. The aspartic protease DDI-1 functions in the SEL-11-dependent ERAD of SLO-1. (A)** Representative images and quantification of SLO-1 at the dorsal cord and body wall muscle. *ddi-1* null mutation and a mutation in the aspartic protease domain of *ddi-1* elevate SLO-1 channels to similar levels. **(B)** Representative images and quantification of SLO-1 at the dorsal cord and body wall muscle. *ddi-1*, *sel-11*, and *ddi-1;sel-11* restore SLO-1 channels to similar levels. Data are means ± SEM; ****P < 0.0001, One-way ANOVA, Tukey's post hoc test. (scale bar = 10 μm).

[27,55]. Interestingly, many of these processing proteins, including SEL-11, SEL-1, and DDI-1, are also necessary for SLO-1 degradation. This raises the possibility that SLO-1 degradation is indirectly regulated by SKN-1A-mediated control of proteasome levels. To test this possibility, we introduced the *skn-1a(mg570)* mutation into *erg-28* animals and determined SLO-1 channel levels. The *skn-1a(mg570)* allele specifically blocks *skn-1a* expression but does not impact other *skn-1* isoform genes [27]. *skn-1a;erg-28* double mutant animals considerably restored SLO-1 levels in both neurons and muscles (Fig 6A), implicating SKN-1A involvement in SLO-1 degradation.

It is possible that *erg-28* mutation causes proteasome dysfunction and the resulting SKN-1A activation enhances SLO-1 degradation. We assessed this possibility by comparing the induction levels of the *rpt-3p::gfp* reporter in wild-type and *erg-28* mutant animals. Compared to wild-type animals, we found no significant difference in either basal or bortezomib-treated conditions (S10 Fig). These data indicate that an *erg-28* mutation is neither a cause of proteasome dysfunction nor a participant in the response to proteasome dysfunction.

Next, we explored the possibility that SKN-1A functions downstream of SEL-11 and DDI-1 for SLO-1 degradation. SKN-1A activation involves N-linked deglycosylation and specific asparagine-to-aspartate substitutions by PNG-1, and N-terminal cleavage by DDI-1 [27,29]. Thus, if SKN-1A functions downstream of SEL-11 and DDI-1 in SLO-1 degradation, then the expression of an *skn-1a* construct that bypasses these post-translational processes, *skn-1a[cut, 4ND]*, would reduce SLO-1 levels. Indeed, expression of *skn-1a[cut, 4ND]* under muscle-specific *myo-3* and pan-neuronal *rgef-1* promoters reduced SLO-1 levels in *ddi-1;sel-11 erg-28* mutants (Fig 6B), indicating that SKN-1A functions downstream of SEL-11 and DDI-1.

Next, we investigated whether the classical ERAD and SKN-1A-mediated degradation pathways function in parallel or in series to degrade SLO-1. We used the miniMos transposon to express a single-copy of *skn-1a[cut, 4ND]* under the pan-neuronal *rgef-1* promoter. This transgene, *cimSi4*, strongly induced the expression of the proteasome reporter *rpt-3p*::GFP in the nervous system without bortezomib treatment, indicating that SKN-1A[cut, 4ND] is indeed constitutively active (Fig 6C). We introduced the *cimSi4* transgene into *sel-11 erg-28* mutant animals and compared the levels of SLO-1 channels with those of *erg-28* and *sel-11 erg-28* animals. Expression of *skn-1a[cut, 4ND]* partially reduced SLO-1 levels in the dorsal cord of *sel-11 erg-28* mutants, but its SLO-1 levels were still significantly higher than those observed in *erg-28* mutants (Fig 6D). Moreover, we found that the introduction of a *png-1(ok1654)* mutation considerably restored SLO-1 levels in *erg-28* mutant animals (S11A and S11B Fig). Together these data indicate that SEL-11 and DDI-1 directly participate in SLO-1 channel degradation and are also involved in SKN-1A processing/activation, which in turn influences SLO-1 degradation by inducing proteasome subunit expression.

## Discussion

In this study, we show that inefficient anterograde transport of SLO-1 channels from the ER causes their degradation by the classical ERAD and proteasomal dysfunction response pathways. We found that classical ERAD proteins remove and degrade SLO-1 channels from the ER membrane. Interestingly, the transcription factor SKN-1A, which is activated by proteasome dysfunction, also contributes to SLO-1 degradation by maintaining proteasome levels.

Our genetic analyses reveal that SLO-1 degradation is a complex process, in which classical ERAD and SKN-1A-mediated proteasomal upregulation arise from their central components: SEL-11 and DDI-1 (S12 Fig). SEL-11, along with SEL-1/HRD3 and CUP-2/DER-2 (Derlin homologs), ubiquitinates and removes SLO-1 channels in the ER. DDI-1 functions downstream of the HRD1 ubiquitin ligase complex to cleave the ubiquitinated SLO-1 channels into

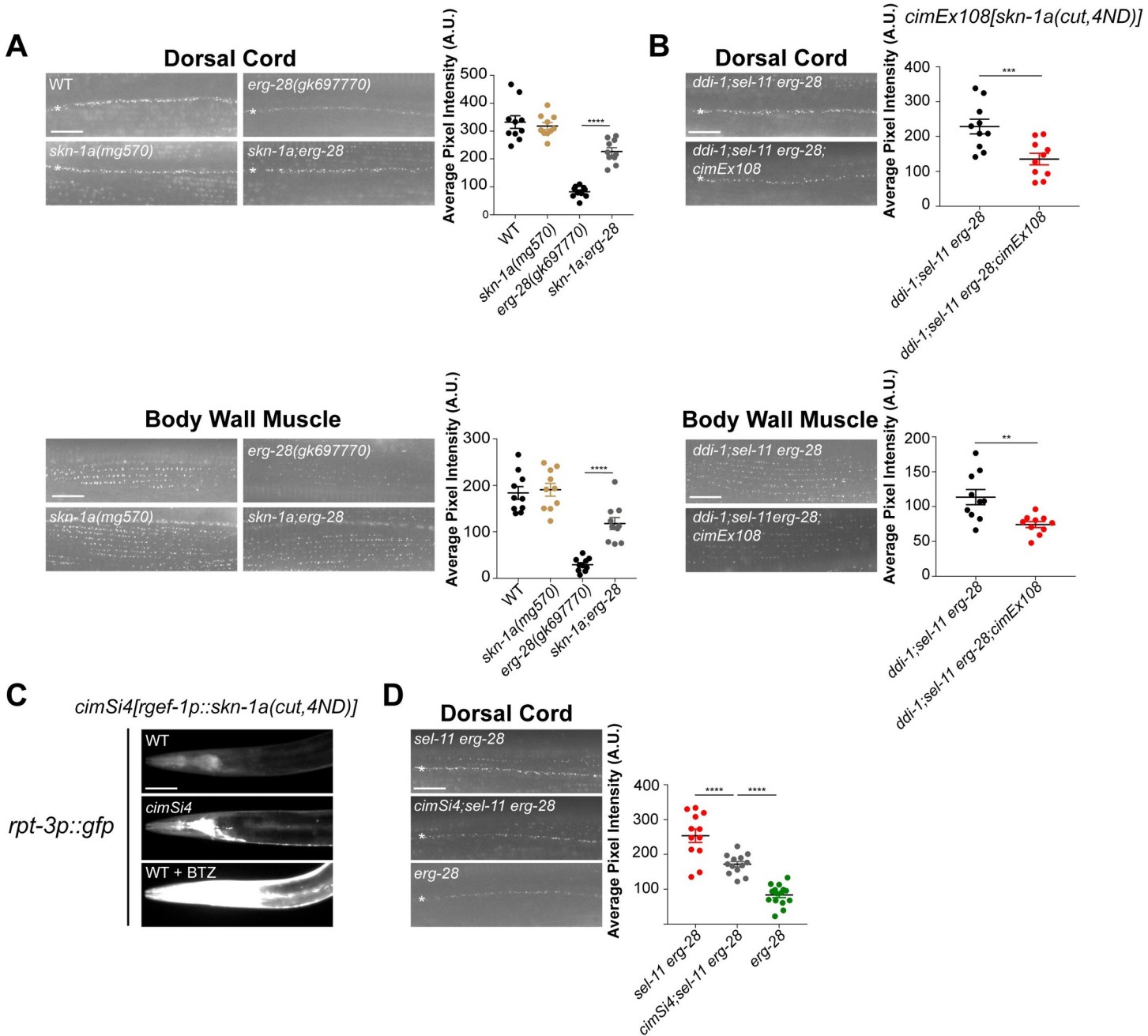

**Fig 6. SKN-1A mediates SLO-1 degradation independently of the SEL-11 ERAD pathway.** Representative images and quantification of SLO-1 at the dorsal cord and body wall muscle. **(A)** *skn-1a* mutation elevates SLO-1 levels in *erg-28* mutants. Data are means ± SEM; ****P < 0.0001, One-way ANOVA, Tukey's post hoc test. **(B)** *cimEx108*, a transgene expressing *skn-1a[cut, 4ND]* under pan-neuronal promoter *rgef-1* and muscle-specific promoter *myo-3*, dampened SLO-1 levels in *ddi-1;sel-11 erg-28* mutants dampened SLO-1 density at the dorsal cord and body wall muscle. Data are means ± SEM; ***P < 0.001; **P < 0.01 (paired two-tailed t-test). **(C)** *cimSi4*, a single-copy transgene of *skn-1a(cut, 4ND)* under the *rgef-1* promoter, constitutively induces the transcriptional *rpt-3p::gfp* reporter (*mgIs72*) expression, specifically in neurons. The bottom panel demonstrates *rpt-3p::gfp* expression when treated with 40 μM bortezomib. **(D)** *cimSi4* partially dampened SLO-1 levels in *sel-11 erg-28* mutants at the dorsal cord, but the remaining SLO-1 levels are still higher than in *erg-28* mutant animals. Data are means ± SEM; ****P < 0.0001, One-way ANOVA, Tukey's post hoc test. (scale bar = 10 μm).

fragments before or after CDC-48.2/p97 unfoldase, which extracts and dislocates SLO-1 channels from the ER membrane. On the other hand, glycosylated SKN-1A escapes SEL-11-dependent degradation and is de-glycosylated and modified by PNG-1 and translocated to the

nucleus, where it is further processed by DDI-1 to induce the expression of genes encoding proteasomal subunits. Thus, *sel-11*, *sel-1*, and *ddi-1* mutations cause both direct SLO-1 degradation and indirect proteasome reduction. Consistent with a previous study [29], our model suggests that not all of the SKN-1A proteins are degraded in the absence of proteasome dysfunction. Rather, a trace amount of SKN-1A proteins becomes active and maintains a normal pool of proteasomes.

Why SLO-1 channels are targeted for degradation by SEL-11-mediated ERAD in the ER? Most of the ERAD substrates are misfolded proteins present in the ER lumen or membrane at a high level. The levels of SLO-1 expression in our study are within a physiological range, since we used CRISPR/Cas9 to fuse the genomic *slo-1* gene with the GFP coding sequence. Furthermore, GFP-tagged SLO-1 channels do not appear to be misfolded in the *erg-28* mutant. This is supported by our observation that *sel-11* mutation considerably restores reduced SLO-1::GFP levels and aldicarb hypersensitivity observed in the *erg-28* mutant (Fig 1 and S3 Fig). One documented example of ERAD activity for a normally folded protein is HMG-CoA reductase. HMG-CoA reductase is degraded by HRD1-mediated ERAD in response to cholesterol (Reviewed in [56]). Interestingly, the structure and topology of HMG-CoA reductase and SLO-1 have some similarities. Both proteins are multi-pass transmembrane proteins with long cytoplasmic tails (approximately 500 amino acids for HMG-CoA reductase and over 700 amino acids for SLO-1) and form a homotetrameric complex. These large cytoplasmic structures may render them as salient SEL-11/HRD1 ERAD substrates.

The physiological advantage of SLO-1 degradation in the ER remains unclear. ERG-28 normally facilitates anterograde SLO-1 trafficking from the ER, and its absence may increase SLO-1 retention time in the ER. The ER is a major storage compartment for calcium ions, and it releases calcium to the cytosol through ryanodine receptors in muscles and neurons. It is possible that the released calcium ions from the ER may bind to ER-retained SLO-1 channels and trigger a conformational change, which is readily recognized and targeted for degradation. This may also explain why an *erg-28* mutation causes lower levels of gain-of-function SLO-1 than those of normal SLO-1, and a combination of *erg-28* and *sel-11* results in comparable levels of both SLO-1 (Fig 1A and S5 Fig). Further investigation is required to understand how ERG-28 protects SLO-1 channels from degradation.

In addition to SEL-11 and SEL-1, the Derlin homologs are also involved in SLO-1 channel degradation. The exact molecular role of Derlin in ERAD has not been clearly defined. *C. elegans* possesses two Derlin homologs, CUP-2 and DER-2. Our data show that both Derlin homologs act redundantly in that only a single homolog is necessary and sufficient to degrade SLO-1 (S6A and S6B Fig). Similar to Derlin, there are two homologs of p97/VCP/CDC48 in *C. elegans*, CDC-48.1, and CDC-48.2, which act to extract and direct ubiquitinated substrates from the ER to the proteasome through an ATP-dependent mechanism [48,57]. Unlike Derlin, we found that *cdc-48.2*, but not *cdc-48.1*, was essential for SLO-1 degradation. These data argue against the conventional idea of functional redundancy between the two homologs [50]. Although they share a high degree of amino acid sequence similarity, CDC-48.2 and CDC-48.1 have distinct developmental expression patterns [58]. How CDC-48.2, but not CDC48.1, can specifically recognize SLO-1 channels needs to be further investigated.

We note that there may be a minor *sel-11*-independent pathway for SLO-1 degradation, evidenced by the incomplete restoration of SLO-1 levels in *sel-11 erg-28* mutants. Since autophagy does not appear to contribute to SLO-1 degradation (S9 Fig), perhaps there is some partial redundancy among the ER-resident E3 ubiquitin ligases. It is possible that a single mutation in the other genes that encode the ER-resident E3 ubiquitin ligases is not sufficient to restore SLO-1 levels. Still, a different combination of mutations with some redundancy may increase SLO-1 levels.

The BK channel is a major negative feedback regulatory mechanism that protects against uncurbed cytosolic calcium rises, and its reduction in density results in calcium dyshomeostasis. This unusual susceptibility of BK channels to ERAD may explain how excitable cells under stressful conditions exhibit calcium dyshomeostasis and pathological calcium signaling, which are a cause of cell death and are often associated with a variety of degenerative diseases.

Proteasome inhibitors, such as bortezomib, are popular anti-cancer drugs. One of their adverse side effects is peripheral neuropathy [59–62]. Often this neuropathic pain is severe enough to terminate treatment. Peripheral nerve endings that mediate pain are known to express voltage-gated calcium channels and BK channels. These channels have been targeted for therapeutic interventions [63,64]. Given that our study indicates the susceptibility of BK channels to proteasome dysfunction, a better understanding of BK channel trafficking may present a promising model to describe the mechanism of neuropathy and identify possible therapeutic targets.

## Methods

### Worm strains and maintenance

All *C. elegans* strains were cultured on NGM (nematode growth medium) plates seeded with *E. coli* OP50 at 20˚C. *C. elegans* strains used are listed in S1 Table.

### Microscopy

Fluorescent microscopy was performed as described previously [21].On day 1 (20–24 hours post L4 stage), adult animals were immobilized on a 2% agarose pad with a 6 mM levamisole solution in M9. Images were acquired with a 63x/1.4 numerical aperture on a Zeiss microscope (Axio-Observer Z1) equipped with a SpectraX LED light engine (Lumencor). Images were captured with Zyla 4.2 plus (Andor) using the same settings (fluorescence intensity, exposure time, and gain) for a given set of data. A line-scanning method in Metamorph (RRID: SCR_002368) was used to quantify the average fluorescence intensity. In each image, a pixel intensity of 150 pixels was measured, followed by subtraction of background intensity from adjacent 150 pixels.

### Western blotting and quantification

Mixed-stage worms were lysed and solubilized in SDS lysis buffer (2% (w/v) SDS, 100 mM NaCl, 10% (v/v) glycerol, and 50 mM Tris HCl, pH 6.8) with sonication. Total protein concentrations of worm lysates were quantified by the Bicinchoninic Acid (BCA) Protein Assay method. On the SDS-PAGE gel, we loaded 200 μg and 150 μg of total protein for the samples from the *slo-1(cim105)* and *slo-1(cim113gf)* animals, respectively. Worm lysates were separated on SDS-PAGE and transferred onto PVDF membranes. The separated proteins were probed with anti-GFP (Cell Signaling Technology, Danvers, MA, #2956, RRID:AB_10828931) or anti-tubulin (Developmental Studies Hybridoma Bank, AA4.3, RRID:AB_579793) antibodies. Pixel intensities of GFP and tubulin bands were quantified using Adobe Photoshop CC 2015 (RRID: SCR_014199).

### Aldicarb-induced paralysis assay

Aldicarb-induced paralysis assay was performed as described previously [21,36]. Aldicarb stock was 100 mM in 70% (v/v) ethanol. NGM agar plates with 1 mM aldicarb were prepared at least one day before the assay. L4 worms were picked 20 hours before the assay. Before the assay, 3 copper rings were placed on the plates, and a drop of OP50 was added in the center of

the rings. After 30 minutes, when the plates were dry, 20 to 30 worms were placed, and their paralysis was examined every 10 minutes. Animals were considered paralyzed when they failed to respond to prodding with a platinum wire. Paralyzed worms were removed from the plates and recorded.

### CRISPR/Cas9 genome editing

The method is based on "co-conversion" [65]. We first pre-assembled Cas9 ribonucleoprotein by combining a *dpy-10* sgRNA, a *slo-1* sgRNA (or *sel-1* sgRNA), and purified Cas9 protein. We then made a mixture that included pre-assembled Cas9 ribonucleoprotein, a repair oligo-nucleotide with the desired point mutation and silent point mutations in the flanking codons, and a *dpy-10* homologous repair oligonucleotide (99 bp). The resulting mixture was directly injected into the gonads of wild-type hermaphrodites. The *dpy-10* sgRNA leads to a DSB, and its aberrant non-homologous end-joining repair leads to the Dpy (dumpy) phenotype in F2 homozygous animals. However, if this DSB is repaired by a homologous repair oligonucleo-tide, which contains a dominant mutation, the resulting F1 animals will exhibit the dominant Rol (roller) phenotype. We picked Rol animals in the F1 and genotyped them for the flanking codon sequence insertion.

### Genetic screen for regulators of SLO-1 degradation

*erg-28(gk69777) slo-1(cim113gf)* animals exhibited a grossly faster and coordinated locomotory phenotype and an obliterated level of SLO-1 in muscles and neurons. Using ethyl methanesul-fonate (EMS), we induced random mutations in germ cells of *erg-28(gk697770) slo-1 (cim113gf)* mutants. The mutagenized worms were grown for two generations to produce homozygous mutants and scored for a slow and sluggish phenotype and then selected for increased SLO-1 levels.

### Bortezomib treatment

Bortezomib (LC Laboratories, MA, B1408) solubilized in DMSO was added to OP50 bacteria seeded NGM agar plates at a final concentration of 20 μM. L4 stage worms were transferred to the bortezomib-containing plates and grown for 24 hours before imaging.

### Locomotory behavior assay

*C.* elegans locomotory behavior assay was performed as described previously [21]. Day one adults (24–30 hours post L4) were placed on NGM agar plates without food. Video frames from three different genotypes were simultaneously acquired from a dissecting microscope fit-ted with GO-3 camera (QImaging) with 500 ms intervals for 2 minutes. Ten animals were used for each genotype, and the experiments were repeated at least 3 times. We calculated the average speed of the tested animals using the Tracking Objects option in Image-Pro Plus 10 (Media Cybernetics). When two animals separate after collision or intersection, new tracks are automatically assigned. This could generate more than 10 tracks for 10 animals.

### Thrash rate assay

Day one adults (24–30 hours post L4) were placed in 35mm NGM agar plates without food with 1.5 ml M9 buffer. We allowed 5 minutes for the worms to adapt to the liquid before recording. Videos were recorded using Ocular (1X lens at 0.63 objective for 30 seconds, 100 ms exposure time with no delay).

## Electrophysiology

Electrophysiological recordings from the *C. elegans* neuromuscular junction were performed in the cut-open worm preparation, as previously described [66]. Evoked postsynaptic currents were recorded from voltage-clamped body wall muscles at a holding potential of -60 mV, after eliciting neurotransmitter release from motor neurons using a stimulating electrode on the ventral nerve cord in the presence of extracellular saline containing 1 mM $Ca^{2+}$.

## Mos insertion microinjection

The synthesized *skn-1a*[cut, 4ND] sequence (IDT Inc) was subcloned under the pan-neuronal *regf-1* promoter into the backbone of the pCFJ151 targeting vector [67], and inserted into ttTi5606. MiniMos transgenic animals were isolated using *unc-119* rescue to select transformants, as described in [67].

## Statistical analysis

We used Prism six to perform statistical analysis. Sample numbers, p-value, and statistical tests are indicated in the figure legends. Sample numbers represent independent biological replicates. All the raw data and statistical analysis summaries are included in the source data file.

## Supporting information

**S1 Fig. Screening of ER-resident E3 ubiquitin ligases for SLO-1 degradation.** Representative images of SLO-1::GFP at the dorsal cords and head muscles, and quantification of SLO-1 at the dorsal cords. **(A)** *rnf-5*, **(B)** *rnf-121*, **(C)** *hrdl-1*, **(D)** *marc-6*, and **(E)** *sel-11* mutations were screened for SLO-1 recovery in the *erg-28* background. *sel-11* mutation showed the most robust SLO-1 recovery. ****P<0.0001, NS, not significant, One-way ANOVA, Tukey's post hoc test. (scale bar = 10 μm).
(TIF)

**S2 Fig. An *erg-28* mutation neither causes ER stress nor is required for tunicamycin-induced ER stress.** Representative images and quantification of the ER stress reporter *hsp-4p*::*gfp* expression in WT and *erg-28* animals when treated with tunicamycin, a known ER stress inducer. Data are means ± SEM; NS, not significant, One-way ANOVA; Tukey's post hoc test). (scale bar = 10 μm).
(TIF)

**S3 Fig. A *sel-11* mutation reverses the reduced SLO-1 function in the absence of ERG-28.** A *sel-11* mutation increases aldicarb resistance in *erg-28* animals. Aldicarb-induced paralysis was analyzed using Kaplan-Meier survival analysis.
(TIF)

**S4 Fig. An *erg-28* mutation does not impede the trafficking of overexpressed SLO-1. (A)** Representative images and quantification of SLO-1 at the dorsal cord of *cimIs10*, a transgene that overexpresses SLO-1::GFP in DA and DB motor neurons. Data are means ± SE; NS, unpaired two-tailed t-test. **(B)** Representative images of SLO-1::GFP accumulated in the ER of both wild-type and *erg-28* mutant animals. No aggregated puncta were observed (scale bar = 10 μm).
(TIF)

**S5 Fig. The *sel-11(tm1743)* deletion mutation recovers higher levels of SLO-1 at the dorsal cord than the *cim54* missense mutation.** Representative images and quantification of SLO-1

at the dorsal cord of *sel-11(tm1743) erg-28* and *sel-11(cim54) erg-28*. Data are means ± SEM; *P<0.05, paired two-tailed t-test. (scale bar = 10 µm).
(TIF)

**S6 Fig. Derlin homologs redundantly target SLO-1 channels for degradation. (A)** and **(B)** Representative images and quantification of SLO-1 at the dorsal cord and body wall muscle. Individual Derlin mutations did not affect SLO-1 levels, but *cup-2;der-2* double mutation elevated SLO-1 levels in an *erg-28* background. Data are means ± SEM; ****P<0.0001; NS, not significant, One-way ANOVA, Tukey's post hoc test. (scale bar = 10 µm).
(TIF)

**S7 Fig. CDC48.2, but not CDC-48.1, participates in the SLO-1 degradation process. (A)** and **(B)** Representative images and quantification of SLO-1 at the dorsal cord and body wall muscle. *cdc-48.2* mutation elevated SLO-1 levels in *erg-28* mutants. **(C)** and **(D)** Representative images and quantification of SLO-1 at the dorsal cord and body wall muscle. *cdc-48.1* mutation did not elevate SLO-1 levels in *erg-28* mutants. Data are means ± SEM; ****P < 0.0001, One-way ANOVA, Tukey's post hoc test. (scale bar = 10 µm).
(TIF)

**S8 Fig. *ddi-1* and *sel-11* genes function upstream of the proteasome. (A)** and **(B)** Representative images and quantification of SLO-1 at the dorsal cord and body wall muscle when treated with 40 µM bortezomib, a proteasome inhibitor. Data are means ± SE; NS, unpaired two-tailed t-test. (scale bar = 10 µm).
(TIF)

**S9 Fig. Autophagy is not required for SLO-1 degradation.** Representative images and quantification of SLO-1 in the dorsal cord (indicated by white asterisk). Data are means ± SEM; NS, not significant, One-way ANOVA, Tukey's post hoc test. (scale bar = 10 µm).
(TIF)

**S10 Fig. A mutation in *erg-28* neither causes proteasome dysfunction nor blocks proteasome dysfunction response.** Representative images and quantification of *rpt-3p::gfp* expression in WT and *erg-28* animals when treated with bortezomib, a proteasome inhibitor. Data are means ± SEM; NS, not significant, One-way ANOVA; Tukey's post hoc test). (scale bar = 10 µm).
(TIF)

**S11 Fig. PNG-1/NGLY1, an essential component of SKN-1A activation, is important for SLO-1 degradation. (A)** and **(B)** Representative images and quantification of SLO-1 at the dorsal cord and body wall muscle. Data are means ± SEM; ****P < 0.0001, One-way ANOVA, Tukey's post hoc test. (scale bar = 10 µm).
(TIF)

**S12 Fig. The model of SLO-1 degradation.** SLO-1 channels are normally trafficked to the Golgi complex with the ER membrane protein ERG-28. In the absence of ERG-28, SLO-1 channels are preferentially targeted for degradation by the SEL-11 E3 ubiquitin ligase complex, which consists of SEL-11/HRD1, SEL-1/HRD3, and Derlin homologs (CUP-2 and DER-2). Ubiquitination of SLO-1 signals extraction from the ER by the CDC-48.2/p97 unfoldase. The aspartic protease DDI-1 cleaves ubiquitinated SLO-1 channels either upstream or downstream of CDC-48.2 to facilitate extraction or proteasomal degradation. Partially degraded SLO-1 channels are ultimately degraded by proteasome. Additionally, a defect in the SEL-11 E3 ubiquitin ligase complex and DDI-1 reduces the overall level of proteasomes by blocking proper

processing of the SKN-1A transcription factor, thus indirectly inhibiting SLO-1 ERAD.
(TIF)

**S1 Table. *C. elegans* strains used.**
(PDF)

**S1 File. This excel file contains the raw data used for all quantitative data figures in Figs 1–6 and all supplementary Figs.**
(ZIP)

## Acknowledgments

We would like to thank Dr. Kelly Oh for helpful discussion, and Drs. Lehrbach and Ruvkun for providing some strains. We also thank the Caenorhabditis Genetic Center, funded by National Institute of Health Office of Research Infrastructure Programs, and the National Bioresource Project for strains.

## Author Contributions

**Conceptualization:** Timothy P. Cheung.

**Data curation:** Timothy P. Cheung, Jun-Yong Choe, Janet E. Richmond, Hongkyun Kim.

**Formal analysis:** Timothy P. Cheung, Jun-Yong Choe, Janet E. Richmond, Hongkyun Kim.

**Funding acquisition:** Hongkyun Kim.

**Investigation:** Timothy P. Cheung, Janet E. Richmond, Hongkyun Kim.

**Supervision:** Hongkyun Kim.

**Writing – original draft:** Timothy P. Cheung, Hongkyun Kim.

**Writing – review & editing:** Timothy P. Cheung, Jun-Yong Choe, Janet E. Richmond, Hongkyun Kim.

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
