## [Decision Letter · Decision Letter 0]

13 Feb 2020

Dear Dr Kim,

Thank you very much for submitting your Research Article entitled 'BK channel density is regulated by endoplasmic reticulum associated degradation and influenced by the SKN-1A/NRF1 transcription factor' to PLOS Genetics. Your manuscript was fully evaluated at the editorial level and by independent peer reviewers. The reviewers appreciated the attention to an important problem, but raised some substantial concerns about the current manuscript. Based on the reviews, we will not be able to accept this version of the manuscript, but we would be willing to review again a version that has been substantially revised. We cannot, of course, promise publication at that time.

Should you decide to revise the manuscript for further consideration here, it is critical that your revisions address the specific points made by each reviewer. We will also require a detailed list of your responses to the review comments and a description of the changes you have made in the manuscript.

If you decide to revise the manuscript for further consideration at PLOS Genetics, please aim to resubmit within the next 60 days, unless it will take extra time to address the concerns of the reviewers, in which case we would appreciate an expected resubmission date by email to plosgenetics@plos.org.

[LINK]

We are sorry that we cannot be more positive about your manuscript at this stage. Please do not hesitate to contact us if you have any concerns or questions.

Yours sincerely,

Anne C. Hart

Associate Editor

PLOS Genetics

Gregory Barsh

Editor-in-Chief

PLOS Genetics

Reviewer's Responses to Questions

**Comments to the Authors:**

Reviewer #1: Cheung et al present a nice study using C. elegans to determine that BK potassium channel density is regulated by ERAD and proteasome degradation via the transcription factor SKN-1. All of the molecules studied have orthologs in mammals. Their epistasis analyses using behavioral, physiological and GFP-tagged molecules in nervous system and muscle present a mostly clear story on how levels of the BK channel SLO-1 appears to be regulated from the ER to the Golgi. Impressively, the authors worked hard to find genes involved in this process via reserve and forward genetics, and along the way they generated many double, triple mutants with many genes inconveniently located on the same chromosome. Additionally, they introduced a variety of difficult-to-generate transgenetic strains. This includes (1) a CRISPR knock-in of a gain-of-function mutation in the endogenous slo-1 gene that was already tagged in frame with gfp, (2) a miniMos knock-in of an activated form of skn-1a that predicted to keep the proteasome extra active. I feel that the writing is mostly clear and compliment them on a well-done study.

Major issues:

1. I’m confused about whether you think that normal, correctly folded SLO-1 is subject to this form of degradation, and/or only misfolded versions of SLO-1. Do you think that the *delay* in trafficking of SLO-1 in the erg-28 mutant itself leads to a misfolding of SLO-1? And therefore, the misfolding might be detected and use SEL-11 to direct it to the proteasome? Do you think that the gf mutant version of SLO-1 chosen is subject to misfolding more than WT version? If so, how might this effect your interpretation of results?

2. Starting on pg 12 and in figure 5, you deduce that ddi-1 and sel-11 work in series to elevate SLO-1::gfp in erg-28 since combining them does not elevate higher than either single mutant. However, how do you know that SLO-1::gfp levels at a ceiling that would prevent you from seeing additive effects here? Same issue for testing together with the proteasome inhibitor bortezomib.

3. The novel single-gene knock-in transgenic worm carrying pan-neuronal cimSi4, Prgef-1::skn-1a[cut, 4ND], seems very useful as a way to upregulate proteasome function. Although the present results are consistent with it upregulating not only proteasome expression but also activity, it would be safe to test this idea independently.

Minor issues:

1. You might consider replacing N2 with WT throughout, expect for Methods, so that people outside of worm field may better understand your work.

2. You talk about a deletion mutant for sel-11 in Fig 1&2 and a distinct sel-11 mutant in Fig 3. This is confusing for readers. To help them understand, note the allele for each in all figures so they know which one is which. Also related, in panel Fig 2e, it is confusing to only note the allele cim54 without letting readers know it is an allele of sel-11. Please note sel-11(cim54) erg-28 here, without a semicolon to be consistent with the sel-11 erg-28 in Fig 1. It is convention to use a space rather than a semicolon for mutations on the same chromosome in C. elegans.

3. The title for Fig2 doesn’t fit with the actual data on behavioral and physiological consequences for suppressing the slo-1(gf) phenotypes. These data do not directly demonstrate that SEL-11 targets SLO-1 for “degradation”. They are consistent with this idea, but do not directly support it. Fig 2a and c, having + to represent slo-1(gf) single mutant is confusing, especially since it looks like an “x” when tilted. Fig 2b. there appears to be a light green line that does not match any of the colors in the key. It is also difficult to determine which line matches the slo-1(gf) single mutant. There has got to be a better coding scheme for these data.

4. In Fig 3a& b, it is hard to be convinced that SLO-1::gfp levels are normal in the gf allele due to saturated image. Perhaps you could include a colorized version of a non-saturated greyscale image to better show this finding? In Fig 3b I’m confused why you need arrows. They don’t signify time or order, so eliminate. Fig 3c seems incomplete since it doesn’t indicate mutagenized worms, and other aspects of the screen. Nowhere in Fig 3 do you describe what transgene is included on cimEx107, so its impossible to interpret Fig3e&f. For Fig 3g, it would be useful to highlight the area deleted by allele tm1743.

5. The photos focusing on the dorsal cord show GFP-tagged SLO-1 in the neurons but also in muscle nearby. Help the reader naïve to worms focus on the nerve cord with a bracket on the sides of all photos.

6. Readers will not understand contents of cimEx108 for Fig 6. Why not replace with [skn-1a]? Or something similar? Likewise, think of a way to write cimSi4 more informatively in the figures; you can probably skip noting the sel-11 and erg-28 alleles to save space here.

7. The figure legend of Fig S1 refers to Fig 1 that sel-11 suppressed decreased expression of SLO-1::gfp in erg-28 more than in other ER-resident E3 ubiquitin ligases, but it would be nice to see the sel-11 results repeated side by side with the sel-11 results in Fig S1 for easier comparison.

8. Line 137. List all ER-resident E3 ubiquitin ligases tested here.

9. Note “in vivo imaging” on line 150. These in vivo results are much more meaningful than any Western blot.

10. Pg 10, line 250. You note the gene der-2; however, Wormbase does not list any gene by this name. Instead, it is currently named R151.6 and noted is an ortholog of human DERL2 (derlin 2). Please list the appropriate gene name so that others can follow along with Wormbase. Or get Wormbase to change the name of the gene before you publish your paper.

Reviewer #2: In the manuscript “BK channel density is regulated by endoplasmic reticulum associated degradation and influenced by the SKN-1A/NRF1 transcription factor”, the authors employ genetic analyses to identify key players in the degradation of the SLO-1 channel in C. elegans. The authors use a candidate gene approach to demonstrate that the E3 ubiquitin ligase, SEL-11, is responsible for SLO-1 degradation in erg-28 mutant animals. Furthermore, using an unbiased forward genetic screen the authors identified sel-11 as a gene involved in SLO-1 degradation. Next, the authors continue to elegantly and carefully test candidate genes in the ERAD degradation pathway and propose a model where the efficiency of ER trafficking balanced with ER degradation affects density of functional SLO-1 levels at the presynaptic axon terminal.

The manuscript is a solid body of work that contributes to our understanding of BK channel density at the cell surface. Alterations in channel density impact a plethora of functions such as synaptic transmission, secretion, and hormone release. Changes in functionality or distribution of these voltage calcium activated potassium channels are associated with multiple diseases such as alcoholism, epilepsy and neurodegenerative diseases. This manuscript provides evidence of the susceptibility of BK channels to ERAD. These findings have important implications for maintaining proper functionality of excitable cells. I found this work to be interesting, well executed, and well communicated and would be appropriate for publication in PLoS Genetics after addressing the issues outlined below.

1. The authors’ classification of ERG-28 as an ER chaperone is overreaching. This classification must be demonstrated by documenting changes in the conformation of SLO-1 in the absence of ERG-28. Turnover by ERAD can be a result of misfolding or in appropriate accumulation. The proper designation for ERG-28 an ER resident protein or ER membrane protein. Not an ER chaperone.

2. The authors state that SLO-1 tagged GFP is normally folded based on the functional data demonstrating normal movement and aldicarb sensitivity (Oh et al 2017). With this in mind, the authors propose a role of ERAD in a normally folded SLO-1. However, the assays used to justify proper folding lack the sensitivity to make this claim. Experimental data (protease sensitivity, propensity for aggregation) is lacking to show folding is “normal”.

Additional

1. In line 135, the authors state that they introduced null mutations in all the known ER-resident E3 ubiquitin ligases. However, they do not provide a reference or explanation as to how these candidates were identified or defined.

2. The authors conclude that SEL-11 is the “major” E3 ubiquitin ligase responsible for SLO-1 degradation in erg-28 mutant animals. Quantitative measures of pixel intensity for each of the other E3 ubiquitin ligase in Figure S1 are needed to demonstrate this conclusion (similar to quantification in Figure 1A and 1B, and in Fig S3-S6).

3. Quantification of SLO-1 levels in the Western in Figure 1C are needed to demonstrate that the sel-11 mutant “considerably restored” SLO-1 levels (Line 152). Is this difference significant?

4. In Figure 2, the color scheme in panel B makes it difficult to identify which curve matches which genotype in the key. Considering using fewer beige/brown/gold colors.

5. In line 209, the authors state that the slo-1(cim113gf) animals show low-frequency thrashing in Figure 3E. However, the figure panel indicates the slo-1(ky399gf) allele is used. This is also the case in Figure 3D (with the GFP signal, should be cim113gf). Please review the figure panels and specify correct allele used in the experiment.

6. Quantification of the levels of SLO-1 rescue by sel-11(cim54) in Figure 3F is recommended. It would be interesting to compare the levels in the missense vs null allele (Figure 1C). This analysis may reveal whether this point mutation affect the overall stability of SEL-11 protein or if this point mutation does indeed impair the interaction with substrate SLO-1.

Reviewer #3: Cheung et al. Manuscript Review

This study shows that in the absence of the ER chaperone ERG-28, the BK channel SLO-1 is degraded via the SEL-11 ligase complex as part of ERAD, and in parallel via a pathway regulated by SKN-1A. More specifically, Cheung et al. show that in erg-28 mutants, SLO-1 protein levels in the dorsal cord and muscle are strongly reduced. This reduction in SLO-1 protein is partially suppressed by mutants lacking components of an ER resident E3 ligase complex including SEL-11/HRD1, SEL-1/HRD3, CUP-2 and DER-2. ERAD is known to be mediated by SEL-11 and by the AAA-ATPase CDC-48. Interestingly, SLO-1 degradation requires CDC-48.2 but not CDC-48.1, which may be the first specific function for CDC-48.2. Based on these observations, the authors conclude that SLO-1 is degraded via ERAD in the absence of erg-28. The authors identified sel-11 as a regulator of SLO-1 based on two independent approaches: (i) a candidate screen of 5 ER resident E3 ligases, and (ii) a clever forward genetic screen that capitalized on the sluggish movement phenotype of a slo-1 gain of function mutant. The authors nicely show that erg-28 mutants suppress slo-1(gf) phenotypes including sluggish movement, resistance to paralysis by aldicarb, and reductions in evoked currents in muscle. This suppression is likely due to ERAD because the suppression of SLO-1 function can be reversed by mutation of the sel-11 ligase. These experiments suggest that the SLO-1(gf) channels are functional even in the absence of the ER chaperone erg-28. A major finding in this manuscript is that normally folded SLO-1 is surprisingly degraded via ERAD. While the genetic experiments are consistent with this idea, they do not provide definitive evidence that SLO-1(gf) channels are folded normally in the absence of the chaperone erg-28. The simplest explanation is that the ER chaperone erg-28 is required for normal folding or tetramer assembly of BK channels and in the absence of erg-28, the unfolded or misassembled SLO-1 subunits are targeted for ERAD. This interpretation is consistent with ERG-28 acting as a traditional chaperone for SLO-1. In this alternative model, inhibition of ERAD of SOL-1(gf) in sel-11 mutants gives misfolded SLO-1(gf) extra time to refold in the ER and to continue along the secretory pathway to the plasma membrane. Some misfolded SLO-1 protein may be refolded in erg-28 sel-11 mutants and continues along the secretory pathway to the plasma membrane and the remaining misfolded SLO-1 is degraded via another SEL-11 independent pathway (maybe dependent on autophagy/lysosomes). The partial restoration of SLO-1 levels in the DNC and muscle in erg-28 sel-11 doubles in Fig 1 are consistent with this model.

Major comments

1. The fact that SLO-1 levels are not fully restored to wild type in sel-11 mutants (Fig 1) suggests that SLO-1 might be degraded via another pathway independent of sel-11. This data is consistent with a model where SLO-1 is misfolded in the absence of the chaperone erg-28 and degraded partly via a sel-11/ERAD mechanism and partly via another sel-11 independent pathway (perhaps via autophagy in the lysosome). Does inhibition of the proteasome and/or lysosome completely block the reduction in SLO-1 levels seen in erg-28 single mutants?

2. A major conclusion of this paper is that in the absence of the ER chaperone ERG-28, SLO-1 is normally folded yet still degraded via ERAD. Although the SLO-1(gf) experiments in Figure 2 suggest that the SLO-1(gf) channels are ultimately functional if ERAD is blocked in sel-11 mutants, these experiments do not definitively show that SLO-1 is normally folded in erg-28 mutants. How do the authors know that SLO-1 is not misfolded or improperly assembled in the absence of erg-28? The authors should tone down their conclusion that normally folded SLO-1 is degraded via ERAD and discuss the alternative that SLO-1 is simply misfolded in the absence of erg-28 and targeted for ERAD.

3. The SLO-1(gf)::GFP levels in erg-28 sel-11(tm1743) mutants should be imaged and quantified in the dorsal cord or muscle to allow comparison to the effects of erg-28 sel-11(tm1743) on SLO-1::GFP in Figure 1. The partial restoration of SLO-1::GFP levels in sel-11 erg-28 mutants in Fig 1 and Fig 3D compared to the almost complete restoration of SLO-1(gf) function in Fig 2 suggests that much lower levels of SLO-1(gf) channels can fully restore function of SLO-1(gf) despite lower levels of expression. These results suggest that SLO-1(gf) function can be restored if some of the channels manage to properly fold/assemble in erg-28;sel-11 double mutants. These results also might suggest that misfolded BK channels that cannot be refolded in erg-28 sel-11 mutants ultimately get targeted for degradation via another sel-11 independent pathway.

4. If endogenous SLO-1 is degraded via ERAD in erg-28 mutants, then erg-28 mutants might be expected to have similar functional defects as slo-1 loss of function mutants. Do erg-28 mutants have functional defects (such as aldicarb or thrashing defects) that are similar to slo-1 loss of function mutants? And are these defects suppressed by sel-11?

Minor comments

1. In Figure S4A and S4B, the cdc-48.2 genotypes under the graphs are incorrectly labeled cdc-48.1

2. In Figure 3A, it is not possible to conclude that SLO-1 is expressed at similar levels in cim105 and cim113gf because the GFP signal in these Nerve Ring images are saturated. Non-saturated images should be shown.

3. Figure 3E is labeled slo-1(ky399gf) above the graph but referred to as slo-1(cim113gf) in the legend. Also, please clarify if the cim113gf point mutation is the same one found in the ky399gf allele.

4. Is SKN1A required for normal expression of proteasome genes? If SKN-1A is only involved in regulating proteasome gene expression after proteasome dysfunction, why does mutation of skn-1A suppress SLO-1 degradation? Does erg-28 mutation cause proteasome dysfunction?

5. Figure 6B suggests that activated SKN1A can induce further degradation of SLO-1 even in the absence of sel-11. It would be helpful if the authors discuss what other degradative pathways might be utilized.

6. Does Bortezomib block the degradation of SLO-1 induced by activated SKN-1A shown in Figure 6B?

8. It would be interesting and surprising if SLO-1 is folded normally in erg-28 mutants but still degraded by ERAD. Although, the function of such a mechanism is not clear. The one example of a normally folded protein being degraded via ERAD cited in the discussion indicates that normally folded HMG-CoA reductase is degraded via ERAD in response to increased levels of cholesterol. The physiological function of degrading normally folded HMG-CoA reductase in response to increased cholesterol makes physiological sense because this is a negative feedback pathway where the cholesterol end product signals to degrade HMG CoA reductase, an enzyme that lies upstream in the biosynthetic pathway. In contrast, the physiological function of degrading normally folded SLO-1 in erg-28 mutants is not clear. The simplest explanation is that in the absence of the ER chaperone ERG-28, SLO-1 is misfolded or misassembled and thus degraded via ERAD. This alternative interpretation should be considered in the discussion.

**Have all data underlying the figures and results presented in the manuscript been provided?**

Reviewer #1: Yes

Reviewer #2: Yes

Reviewer #3: Yes

PLOS authors have the option to publish the peer review history of their article (what does this mean?). If published, this will include your full peer review and any attached files.

Reviewer #1: No

Reviewer #2: No

Reviewer #3: No

---

## [Decision Letter · Decision Letter 1]

5 May 2020

Dear Dr Kim,

We are pleased to inform you that your manuscript entitled "BK channel density is regulated by endoplasmic reticulum associated degradation and influenced by the SKN-1A/NRF1 transcription factor" has been editorially accepted for publication in PLOS Genetics. Congratulations!

Yours sincerely,

Anne C. Hart

Associate Editor

PLOS Genetics

Gregory Barsh

Editor-in-Chief

PLOS Genetics

Comments from the reviewers (if applicable):

Reviewer's Responses to Questions

**Comments to the Authors:**

Reviewer #1: The authors have revised their manuscript to satisfy my concerns and produce an interesting and thorough study.

Reviewer #2: Having examined the author's reply to my comments, and to the comments

of other referees, I am fully satisfied and recommend publication.

Reviewer #3: All my concerns have been addressed in the revised version of this manuscript.

**Have all data underlying the figures and results presented in the manuscript been provided?**

Reviewer #1: Yes

Reviewer #2: Yes

Reviewer #3: Yes

PLOS authors have the option to publish the peer review history of their article (what does this mean?). If published, this will include your full peer review and any attached files.

Reviewer #1: No

Reviewer #2: No

Reviewer #3: No

**Data Deposition**

http://datadryad.org/submit?journalID=pgenetics&manu=PGENETICS-D-20-00044R1

**Press Queries**

---

## [Editor Report · Acceptance letter]

29 May 2020

PGENETICS-D-20-00044R1 

BK channel density is regulated by endoplasmic reticulum associated degradation and influenced by the SKN-1A/NRF1 transcription factor 

Dear Dr Kim, 

We are pleased to inform you that your manuscript entitled "BK channel density is regulated by endoplasmic reticulum associated degradation and influenced by the SKN-1A/NRF1 transcription factor" has been formally accepted for publication in PLOS Genetics! Your manuscript is now with our production department and you will be notified of the publication date in due course.

With kind regards,

Laura Mallard

PLOS Genetics

On behalf of:
